# Current Advancements in Spinal Cord Injury Research—Glial Scar Formation and Neural Regeneration

**DOI:** 10.3390/cells12060853

**Published:** 2023-03-09

**Authors:** Tanner Clifford, Zachary Finkel, Brianna Rodriguez, Adelina Joseph, Li Cai

**Affiliations:** Department of Biomedical Engineering, Rutgers University, 599 Taylor Road, Piscataway, NJ 08854, USA

**Keywords:** spinal cord, traumatic injury, glial scar formation, neural regeneration, therapy, cell transplantation, cell reprogramming, neural stem progenitor cells

## Abstract

Spinal cord injury (SCI) is a complex tissue injury resulting in permanent and degenerating damage to the central nervous system (CNS). Detrimental cellular processes occur after SCI, including axonal degeneration, neuronal loss, neuroinflammation, reactive gliosis, and scar formation. The glial scar border forms to segregate the neural lesion and isolate spreading inflammation, reactive oxygen species, and excitotoxicity at the injury epicenter to preserve surrounding healthy tissue. The scar border is a physicochemical barrier composed of elongated astrocytes, fibroblasts, and microglia secreting chondroitin sulfate proteoglycans, collogen, and the dense extra-cellular matrix. While this physiological response preserves viable neural tissue, it is also detrimental to regeneration. To overcome negative outcomes associated with scar formation, therapeutic strategies have been developed: the prevention of scar formation, the resolution of the developed scar, cell transplantation into the lesion, and endogenous cell reprogramming. This review focuses on cellular/molecular aspects of glial scar formation, and discusses advantages and disadvantages of strategies to promote regeneration after SCI.

## 1. Introduction

Spinal cord injury (SCI) is a debilitating affliction and results in a wide range of physical deficits, e.g., motor, sensory, and autonomic. Deficits include chronic pain, loss of bladder control, respiratory system strain, and loss of motor function causing immobility below the injury level. Recent studies have estimated that the overall global prevalence of SCI is 20.6 million cases and 250,000 to 500,000 patients each year suffer from SCI [1,2]. After SCI, a tissue scar forms surrounding the injury epicenter composed of glial and supporting cell types. However, many of these invading cell types also contribute to an inflamed, inhibitory microenvironment detrimental to neural regeneration [3]. This inhibitory microenvironment suppresses neural regeneration via secreted molecules that inhibit neuronal function or prevent axogenesis, e.g., chondroitin sulfate proteoglycans (CSPGs), Nogo-A, and myelin-associated glycoprotein (MAG) [4,5,6]. To enhance neural regeneration, the glial scar may be manipulated to reduce its negative consequences or synergistically enhance positive qualities. This review discusses cellular glial scar formation and recent advancements in cellular/molecular-based treatments to promote neural regeneration. 

### 1.1. Cellular Events Immediately following Trauma

Immediately, the physical impact causes ischemia, mechanical damage, and physical ruptures in cell processes, organelles, and membranes [7]. Ischemia, in addition to damage-mediated ion channel defects and rapid calcium release via cell lysis, contributes to ionic imbalance at the injury epicenter [8,9]. Due to the mechanical damage, neurons often lose their function via axonal lesions and axons degrade and retract toward the soma, a process known as Wallerian degeneration [10,11]. Locally secreted inhibitory and inflammatory signals induce chemorepulsion of the axonal growth cone, resulting in an inability to generate new axons in the immediate area of the injury. Chronic demyelination of axons occurs primarily at 24 h post-injury and remains prevalent up to one year post-injury [12], a process by which immune cells continuously attack newly synthesized myelin, inhibiting neuronal function. In conjunction, oligodendrocytes lose their function due to mechanical processes and somal cleavage resulting in an inhibitory environment for functional axon formation. Myelin-associated molecules are dispersed around the injury site from injured axons and oligodendrocytes. These molecules, primarily MAG, oligodendrocyte myelin glycoprotein (OMgp), and Nogo-A, are all known inhibitors of neural, axonal regeneration, and plasticity, and persist during scar formation and maturation [4,6,13]. Surviving cells attempt remyelination, but recovery is chronically diminished due to the inhibitory microenvironment and physical barrier generated by astrocytes and supporting scar border cells [6]. A cascade of cellular/molecular events occur resulting in mass cell death, spreading inflammation, and an inhibitory microenvironment [14,15,16]. Cell membranes are lysed during the injury impact, inducing a spike of depolarization in local cells. The depolarization of cell membranes and the dysregulation of homeostatic processes results in the release of many different types of cell signals [16]. This local damage signaling cascade and ischemia-initiated damage results in mass ATP release by cells and serves as the primary signaling molecule to initiate glial scar formation [17,18,19]. P2 receptors are present on oligodendrocytes, astrocytes, and oligodendrocyte progenitor cells (OPCs), inducing the first instances of mobility around the injury lesion [20,21]. These initial cellular events also lead to the expression of a variety of signals, e.g., proapoptotic, necroptotic, ferroptotic, proinflammatory, anti-inflammatory, and many signals that serve cell-recruiting functions [22,23,24]. Chemokines, cytokines, alarmins, and damage-associated molecular patterns (DAMPs) initiate the reactive response in cellular components of the scar and define the pathology of glial scar formation via the recruitment of various cell types, e.g., astrocytes, fibroblasts, microglia, neural stem/progenitor cells (NSPCs), fibroblasts, macrophages, and invading immune cells [3,16]. 

### 1.2. Glial Scar Formation 

The glial scar is composed of various cell types: astrocytes, fibroblasts, NSPCs, microglia, macrophages, and immune cells (Figure 1). Glial scar formation is induced by a combination of cell signals to the surrounding area following injury and develops for months after SCI [25,26,27,28,29]. These signals induce a heterogenous population of glial cells, primarily resident astrocytes, into a state of reactive gliosis. During reactive gliosis, astrocytes become activated and exhibit major differences in soma size, cell location, morphology, enhanced proliferation, and transcriptional profile [21,30,31,32]. Astrocytes translocate and congregate around the lesion site while expressing filament proteins such as glial fibrillary acidic protein (GFAP), Nestin, and Vimentin to stabilize the newly formed astrocytic structure [5,30]. Transposed astrocytes proliferate to form a physical barrier, a process mediated by STAT-3 signaling and leucine zipper-bearing kinase (LZK) expression [25,33,34]. Reactive astrocytes secrete molecular signals, including the yes-associated protein (YAP), CCL2, and Csf2, that contribute to the recruitment of other cell types, primarily immune cells, while physically ensnaring fibroblasts by 7 days post injury (dpi) [35,36,37,38]. Fibroblasts and astrocytes produce extracellular matrix (ECM) molecules to stabilize the glial scar, e.g., CSPGs, fibronectin, laminin, collagen, and proteoglycans from fibroblasts, and provide shape and stability [5,30,39,40,41]. After 1–2 weeks, astrocytic proliferation stops, effectively representing glial scar maturation. A phase change to the chronic state is also emphasized by the glial process shift from perpendicular to parallel and fosters stable and compact tissue formation mediated by STAT-3 signaling [25,34,42].

Fibroblast populations are present in the basal laminae and parenchyma of the brain and are distributed consistently throughout the spinal cord. After SCI, fibroblasts become activated to proliferate and migrate toward the lesion by 3 dpi [38,43,44]. This cell population is bolstered by nearby pericytes which undergo differentiation into stromal fibroblasts, a process which peaks at 2-weeks post-injury [29,38]. The primary function of fibroblasts after SCI is to produce stromal ECM molecules and fortify the glial scar structure, as well as provide inhibitory signaling to generate a physicochemical barrier that protects the injured tissue from invading species. This molecular activity results in fibrosis and the formation of a clear “fibrotic scar” medial to the glial layer, composed of fibroblasts and surrounding stromal ECM [29]. In clinically relevant contusion SCI, fibroblast deposition around the injury site increases dramatically by 5 dpi and peaks at 7 dpi. In contrast, in dorsal hemisection SCI, fibroblast deposition peaks at 9–14 dpi [45]. The fibrotic aspect of the SCI scar is seen as an absolute barrier to neural regeneration and is a key motivator for many therapeutic approaches [46].

Native and invading immune cells contribute to inflammatory statis, a defining aspect of the glial scar’s pathology. Microglia constitute the primary immune response to general CNS injury and work in tandem with macrophages as the major reactive cell type immediately after SCI [38]. The roles of microglia and macrophages often overlap due to their common lineage, morphological characteristics, molecular protein markers, and genes expressed after injury. It is accepted, however, that microglia subtypes perform a diverse variety of functions in the healthy and injured spinal cord [3,47,48]. These processes include reactive phagocytosis, the recruitment of immune cells, antigen processing, and the regulation of the pro-inflammatory condition [16]. Within 24 h, the majority of immune cell types are recruited to the injury site, including monocyte-derived or microglia-derived macrophages, leukocytes, T cells, and B cells [3]. These populations induce reactivity and a pro-inflammatory state around the glial scar via DAMPs and inflammatory cytokines [47]. Microglia possess the lowest threshold for reactivity and initiate glial scar formation via IGF-1 release stimulating astrocyte reactivity and proliferation, as well as improving scar formation itself via the downregulation of astrocytic P2Y1 receptors [17,18,21,47]. Microglia persist around the injury site for up to 6 months and contribute to the chronic state and neuropathic pain [47]. Microglia also maintain cellular crosstalk with the major cell types in the glial scar and are thus a vital cell type to gain understanding of for SCI therapeutic development [21,48,49,50,51,52].

Macrophages play an integral role in scar development and recovery by performing a variety of functions including notable phagocytosis of cell debris around the injury site [48,52]. Monocytes are recruited to the injury by astrocytes during the acute phase via chemokine/cytokine release, e.g., CCL2, CCL5, and CXCL8 [53,54,55], and differentiate into macrophage subtypes: pro-inflammatory (M1) and anti-inflammatory (M2). Astrocytes induce macrophage polarization and mobility toward the lesion via chemotaxis [53]. In the subacute SCI phase, M1 macrophages upregulate astrocyte activation to induce glial scar formation while M2 macrophages secrete TGF-β in vitro [56,57]. Crosstalk between macrophages and other common glial-scar-forming cell types allows M2 macrophages to polarize astrocytes and direct glial scar contributions [55,57]. Macrophages and fibroblasts migrate to the neural lesion and contribute to scar formation up to 14 dpi. However, after macrophages are cleared from the scar, the density of the fibrotic region is significantly reduced/disrupted, suggesting a role of macrophages to manipulate fibroblast migration and position [38,58]. 

Immediately after SCI, NSPC populations transition from quiescent to the activated state, e.g., neural stem cells, ependymal progenitors, and OPCs/NG2 polydendrocytes [59]. Activated NSPCs proliferate and differentiate into glial lineage cells and contribute to the formation of the glial scar border [22,27,28]. Multipotent asymmetric division allows NSPCs and progeny to differentiate into the glial cells of the scar border in synchrony with recruited resident astrocytes. Ependymal cells have been shown to possess stem-like qualities after neural injury such as self-renewal and multipotency [60,61]. Ependymal cells line the central canal of the spinal cord and guide adjacent cerebrospinal fluid (CSF) via cilia into the lateral ventricles of the brain. Ependymal cells contribute to few clinical injury types due to low migratory capacity even in an activated state. However, in stab and contusion SCI, ependymal cells become activated, revert to a stem-cell-like state, and contribute to astrocyte population due to central canal damage [59]. Ependymal progenitors are thus a popular target for regenerative therapies. 

Oligodendrocyte progenitor cells (OPCs) differentiate into oligodendrocytes and Schwann cells to remyelinate neurons and repair damaged sheaths [26,27]. OPCs also express NG2, a proteoglycan responsible for the trapping, congregation, and efficient myelination of many neurons [28,37]. NG2+ polydendrocytes are a population containing OPCs, macrophages, and pericytes displaying stem-cell-like characteristics of self-renewal and multipotency before and after SCI [62,63,64]. This NSPC population is spatially distributed in a checker-like pattern throughout the mammalian spinal cord. Thus, this population contributes to the glial scar of various SCI types and grades due to distribution and migratory capacity [59]. NG2 polydendrocytes natively possess qualities of their differentiated progeny, e.g., hypertrophy and secretion of inhibitory molecules that inhibit axogenesis. Furthermore, this population can form functional synapses with neurons in the immediate vicinity. NG2 polydendrocytes provide a potential target for therapeutics due to NSPC qualities and interactions with crucial cell types for synaptic transmission and glial scar formation [37,62].

### 1.3. Positive and Negative Effects of the Glial Scar

The nature of the glial scar in the context of regeneration after SCI is highly debated in the field [36,65,66]. While the scar may inhibit regeneration through the lesion, it also preserves the viable tissue surrounding the lesion [67]. Glial scar formation results from a combination of complex processes including inflammation, reactive gliosis, apoptosis, autophagy, and others [68,69,70]. To design effective therapeutics, the potential consequences of manipulation of each process must be carefully considered. In the field, two strategies to target the glial scar have been developed: targeting glial scar formation and targeting aspects of the established scar. Positive and negative effects of the glial scar in the injured spinal cord have been extensively explored (Table 1). Collectively, these observations have fostered the development of research strategies to target scar formation or components of the established scar as the next stage of SCI therapeutic development. Moreover, a combinatorial approach may be the most effective to establish a treatment that is optimal.

## 2. Current Research in Therapeutic Interventions

Many therapeutic approaches have been developed to treat SCI; however, due to the heterogeneity and complex pathophysiology, no effective treatment options exist. Treatment strategies can be categorized as follows: behavioral, biological, device, drug, dietary, procedural, radiation, and combinatorial. In the following sections, current advancements in biological treatments are detailed with a focus on the glial scar as an inhibitory barrier. Biological strategies to treat SCI include the targeting of scar formation, the resolving of the glial scar for regeneration, cell transplantation methods, and endogenous cell reprogramming (Figure 2). Each treatment strategy implements a separate philosophy for the mode of action and aspects of innate recovery to be maintained. These methods also vary greatly in clinical successes and feasibility, crucial variables to consider for a successful SCI therapeutic. 

### 2.1. Therapeutics Targeting the Glial Scar

#### 2.1.1. Targeting Scar Formation

The glial scar presents a physicochemical barrier to regeneration; thus, hindering glial scar formation or decreasing the aggregation of astrocytes may reduce or remove this barrier to promote regeneration [34]. Major approaches include the inhibition of the signal transducer and activator of transcription 3 (STAT3), transforming the growth factor beta (TGF-β) signaling pathway, and engineering neurons for better survival in the injury microenvironment. The primary approach of targeting aggregation is to inhibit STAT3, a key growth factor for cell proliferation in the reactive state and transcription factor phosphorylated and activated by Janus kinase (JAK) [34]. Inhibiting these molecular pathways reduced astrocyte proliferation and reduced the glial scar; however, the results are not consistently replicable and yielded no functional recovery [82]. This pathway also exists in microglia, yielding similar results [83]. An alternate approach is the inhibition of astrocyte proliferation via the TGF-β signaling pathway. Two groups have targeted TGF-β directly, upstream elements and downstream elements, but both of which have yielded unclear results due to compensatory mechanisms, resulting from complex overlapping Smad signaling [84,85]. Overall, astrocytes are resilient and maintain a proliferative state regardless of either intervention; thus, this approach may not yield a prospective/successful option for treatment.

Fibroblasts form an absolute barrier to cellular/axonal penetration and prevent neural regeneration [43,46]. They are derived from pericytes, offering a target to intervene in their deposition as the glial scar forms [44]. Pericyte-derived fibroblast deposition was limited by the conditional knockout of pericyte subclass proliferation, resulting in some functional sensorimotor recovery. Functional axons also integrate into the lesion circuitry below the injury [86]. Other observations demonstrate that this subpopulation does not contribute to the glial scar in a stab injury model using cellular fate-mapping and does not have a clear cellular mechanism for recovery [87]. The mesenchymal and fibrotic aspects of the scar can be targeted via genetic manipulations or pharmacological agents to directly disrupt their proliferation. The administration of anti-mitotic drugs Taxol and Epothilone B resulted in microtubule rearrangement, shutting down mitosis and the migration of new fibroblasts to the injury site, and resulted in axon regeneration and functional recovery [88]. Alternative methods include the inhibition of stromal components, e.g., TGF-β-fibrinogen signaling and collagen synthesis, due to their role in the inhibition of axonal growth signals and spatial limitations. This resulted in axon regeneration and neuroprotective qualities, but no functional recovery [78,89]. 

#### 2.1.2. Resolving the Glial Scar

The mature glial scar can also be targeted at a variety of time points, representative of the variable clinical injury and patient population. Neuronal function is hindered by the mature glial scar: (1) cells are damaged and apoptotic; (2) axons are cleaved and degenerate to the soma; (3) inhibitor molecules prevent endogenous neuronal functional and exogenous signals from stimulating function; and (4) neurons are spatially limited by stromal molecules and the fibrotic scar. The main focus of this approach is neuronal regeneration and survival, as the glial scar is a known inhibitor of axogenesis, and thus targeting a variety of glial scar components, as they make excellent candidates for therapeutic strategies [77,90]. 

CSPGs form a perineural net around sprouting axons that inhibits formation in aberrant regions [90]. These nets are degraded by chondroitinase-ABC (chABC), signaling the establishment of mature neuronal circuitry [91,92]. After SCI, CSPGs are released by proliferating astrocytes and fibroblasts and inhibit local cells’ function to protect damaged tissue and maintain the inflammatory state [92]. Thus, the inhibition of the inhibitory stromal ECM molecule function of axon growth has been employed. Periostin contributes to scar formation through inflammatory signaling and fibrosis. The administration of periostin antibodies up to 2 weeks reduced scar pathology and resulted in functional improvements in sensorimotor tasks following mouse contusion SCI [43]. The administration of N-cadherin antibodies blocked its interaction with an integrin, and led to rapid behavioral recovery; however, the mechanism is not understood [42]. CSPGs are ubiquitous and abundant around the scar and possess a known degradation enzyme [91,93,94]. The upregulation or administration of CSPG enzyme ChABC, a bacterial lyase which degrades CSPG side chains, yielded functional recovery in a rodent SCI model [92,95]. However, the high enzyme degradation rate (within 24 h) was considered to be a major limiting factor preceding clinical trials, thus methods of biotechnology are being employed to extend the stability of the enzyme around the injury site. Kosuri et al. developed a machine-learning-based system to synthesize polymers for enzyme stabilization via a directed evolution approach. This system uses active learning to guide copolymer synthesis, defined by chain length and composition, and the analysis of enzyme–copolymer complex thermostability. Three iterative cycles yielded the successful stabilization of the ChABC–copolymer complex at 30 percent activity after one week. In comparison with current approaches, this is the most successful method to stabilize ChABC and the complex should be examined in vivo for the treatment of SCI and the long-term degradation of the glial scar border [96]. The targeting of mature glial scar cells yields few positive results, and the engineering of neurons to survive the negative microenvironment and penetrate through the scar may be more effective. Neurons have been engineered to survive the inhibitory environment during scar formation to determine whether axonal projections can still be made through the lesion and attenuate damage [94]. Neurons with conditionally knocked out CSPG receptors have resulted in some motor and urinary function recovery but they did not translate to humans in clinical trials [93]. Instead of reducing the impact of the inhibitory environment on the cells, attention has also been placed on inducing the desired output directly, via induced axogenesis [97,98,99]. Cell cycle targets resulting in apoptosis have been knocked out or overexpressed to assess their role in neuroprotection. The inhibition of factors such as CDK4/6, NRF2, and Myc which play roles in apoptosis may prevent apoptosis in the cells and induce neuroprotection [68,100,101]. Cell cycle arrest has recently shown promising results; however, the mechanisms of these therapies are not well characterized. Implementing this strategy with the advancing understanding of genetic engineering will enhance SCI therapeutics to facilitate functional recovery.

### 2.2. Cell Transplantation

The glial scar is inhibitory to axon growth and penetration; thus, the manual transplantation of adult cells has been employed to repopulate the neural lesion and promote regeneration after SCI. Neurons are post-mitotic, or terminally differentiated; therefore, the transplantation of cells may be beneficial to supplement the neuronal number or recover native neurons in the injury site. Cell and gene therapies have emerged as popular treatment strategies in the recent decade with new innovations for cell transplantable therapies such as the variety of FDA-approved Car-T cell therapies. Many protocols have been established to isolate pools of cells to be generated for transplantation, e.g., embryonic stem cells (ESCs), NSPCs, or induced pluripotent stem cells (iPSCs). The transplantation of supplementary or maintenance cell types to the neuron population may provide positive effects shared with newly transplanted neurons, e.g., 1. changing the local microenvironment may attenuate inhibition to allow for axonal regeneration, 2. nutritional support and stabilization/recovery of damaged cells, 3. neuroprotection, and 4. the regulation of neuroinflammatory levels to a more beneficial state [22,102,103,104]. Alternative cell types include oligodendrocyte progenitor cells (OPCs), Schwann cells, olfactory nerve sheath cells, brain-derived neurotrophic factor-expressing and neurotrophin 3-expressing fibroblasts, and bone marrow mesenchymal stem cells. The wide variety of cell type candidates allows for many angles to approach the development of therapeutics.

The transplantation of unmanipulated ESCs and NSPCs into rodent models historically resulted in little functional recovery and no functional recovery in human clinical trials, as well as posed ethical and safety concerns such as tumorigenesis due to undifferentiated cell populations [105]. To mitigate this, in vitro methods were developed to differentiate cells before transplantation utilizing transcription factors, e.g., NeuroD, NeuN, and Map2, reprogrammed into neurons for transplantation as neural grafts. The Tuszynski group demonstrated that rodent NSPC transplantation with a number of neurotrophic factors promotes survival, differentiation into mature neurons, and functional synapse formation in full transection rodent SCI [106,107]. Fischer et al. transplanted rodent neuronal and glial progenitor cells into a rodent SCI model and restored bladder and motor function. In addition, lentiviral delivery of neurotrophic factors promoted axogenesis up to 9 mm around the injury site [108]. The Tuszynski group produced the golden standard protocol for cell transplantation into a rodent injury model, a highly clinically relevant cervical contusion injury model [109]. They demonstrated similar results to their previous work translated into this landmark model with rodent NSPCs [110]. Controversially, human ESCs were transplanted into non-human primates to demonstrate high translatability. These human cells survived for at least 9 months post injury under immunosuppression, forming mature synapses and improved forelimb function, serving as a powerful preclinical treatment [111]. The transplantation of an oligodendrocyte progenitor cell line made from human embryonic stem cells (hESCs) into the cervical region of SCI patients in a phase 1 clinical trial resulted in reduced cavity/lesion areas, and limited motor recovery. The use of human ESCs may be limited due to ethical concerns however, and while the transplantation of these cell types may seem favorable, immunorejection by the host is a major issue. Patients receiving this treatment would be immunocompromised indefinitely with a reduced quality of life and increased risk of infection.

In 2006, Takahashi et al. established a method to dedifferentiate and redifferentiate human primary cells [112]. A patient’s own cell sample, commonly skin fibroblasts, can be reprogrammed into stem-cell-like cells, or induced pluripotent stem cells (iPSCs), via a transcription factor cocktail, e.g., Oct3/4, Sox2, Klf4, and c-Myc [113]. This finding resolves immunorejection concerns, and allows for the most translational in vitro models for spinal cord injury treatment [114]. Using these iPSCs as a transplantation cell pool, they can be undifferentiated to form a natural array of cell types which can bolster overall tissue performance or they can be differentiated into specific types, e.g., a neuronal sheath, for grafting into an injury site. Many groups have shown that human iPSC transplants can yield some functional recovery in rodent models. The Tuszynski group transplanted human iPSCs into non-human primates, showing some forelimb functional recovery but only to a minor extent [115]. While some results have been seen, very little to no results have been seen from neural stem cell transplantation in human clinical trials so far, and a more deliberate mechanism for functional recovery is yet to be investigated in successful treatments. 

One key issue with transplanted cells is the minor levels of functional synapses generated, which can only lead to minor levels of function returning in subjects. Few functional synapses have been reported which are necessary to form functional neural circuits, the primary cellular basis supporting functional recovery [116]. However, Linaro et al. showed functional stimulation and integration into the visual cortex circuitry with xenotransplanted human ESC-derived cortical neurons in the presence of egtazic acid, an example template for studies into the spinal cord to probe for functional circuit integration [117]. Ceto et al. showed that NSC grafts can integrate into spinal cord circuitry after injury [118]. To further assist in neural circuit integration, it has been hypothesized that specific long-term stimulation may be ideal to increase synapse formation in transplants. Hideyuki Okano’s lab is leading the field as much greater levels of functional recovery were seen in human iPSC transplants in rodents with food-induced calcium signaling due to clozapine N-oxide administration and the stimulation of virally delivered designer receptors exclusively activated by designer drugs (DREADDs) [119]. Kitagawa et al. showed the function of these DREADDs carrying iPSCs as the mechanism for functional recovery [120]. These receptors have enhanced calcium signaling induced over days after the injury, yielding much greater numbers of functional synapses. Given these results, transplanted cells will most likely require periodic stimulation to assist in their cell–cell-signaling-based integration properties in allowing for functional synapses. DREADDs and other forms of highly specific selective stimulation serve as potential clinical interventions to maintain therapeutic stimulation.

Most cell transplantation therapies aiming to supplement neuronal counts at the injury site have not yielded significant results in clinical trials. We have highlighted relevant currently active or completed clinical trials testing cell transplantation methods to treat SCI and improve functional recovery (Table 2). The use of stem cell types non-specific to neuronal fates may provide integral qualities to induce neuroprotection via paracrine effects and regeneration. Bellak et al. transplanted undifferentiated iPSCs into the rat SCI model which resulted in significant functional recovery through the generation of neuronal and other beneficial cell types. These grafts also expressed glial-derived neurotrophic factor (GDNF) and IL-10, two molecules known to possess neuroprotective qualities and enhance motor function. Cell transplantation methods considering the injury cellular/molecular environment have shown some success in the clinical transplantation of OPCs to provide trophic support and induce myelination, such as the Geron clinical trial [121,122]. The trophic support and paracrine signaling provided by cell types that are not intended to differentiate only or mostly neuronally can attenuate many negative qualities associated with the injury such as demyelination, inflammation, and inhibition of tissue repair mechanisms. Moreover, mesenchymal stem cells offer many useful qualities such as rapid proliferation, homing toward the injury site, a wide range of differentiation possibilities, little to no immune response, ease of extraction and preservation, and no ethical concerns [123,124]. The secretome of MSCs offers a wide range of cellular targets around the injury site for paracrine signaling, leading to anti-inflammatory, immunomodulatory, neuroprotective, and neurotrophic signaling [125,126,127,128]. Finally, MSC transplantation can also induce angiogenesis, a process crucial to wound healing [129,130]. However, some issues do arise with MSC-based therapy results. When translating to clinical trials, MSC-based therapies accel at reaching the clinic but have yet to produce dramatic results. Most results were reported as being confined to those with mild injuries or showed no improvement [131,132,133,134,135]. This has been postulated as being due to inconsistencies with patient injuries, and the lack of neuronal function, even in MSCs showing neuronal antigens [136]. Cofano et al. provides a comprehensive analysis of the potential of MSCs as an SCI cell therapy [123]. 

An avenue to consider for optimism moving forward is the movement toward combinatorial strategies with synergistic elements for bolstered treatment potency. For example, the combination of transplanted cells on biomaterial-supported cell seeding may enhance the effectiveness of the stem cells to properly differentiate and integrate [137]. Scaffolds, hydrogels, and nanoparticles can all provide enhancements to the cells that cannot be added from viral delivery [138,139]. In particular, scaffolds work in conjunction with hydrogels and nanoparticles to provide compartmentalized chambers optimized for cell seating, viability, and axonal branching, facilitating axogenesis over the neural lesion [36,139,140]. They also form physical matrices for cells to adhere to, enhancing tissue aggregation and cell migration which can drastically change the behavior of transplanted or grafted cells, changing differentiation possibilities [141]. These tools are being implemented more as biotechnology improves to provide enhanced results that better translate to humans. Many treatments do not address all features of the injury and combinatorial treatments allow for a greater effect as more concerns are addressed. Transplanted biological and biotechnological materials serve as excellent agents to combine with viral gene therapies, a method that may prove to be the most comprehensive form of SCI therapeutics [142].

### 2.3. Endogenous Cell Reprogramming

Gene delivery has emerged as an effective approach to promote regeneration and reduce glial scar formation via endogenous cell reprogramming in the injured spinal cord [143]. Specific cell or population targeting is necessary to upregulate the gene of interest into appropriate cell types and avoid off-target or adverse effects. Two major approaches to target specific cell types include viral serotypes such as AAV1-9 to preferentially target populations of cells, and cell-specific promoters, which initiate cell-specific gene expression, e.g., the GFAP promoter to target astrocytes in the glial scar border and Nestin promoter to target NSPCs contributing progeny to the glial scar [144]. Thus, gene therapy is a versatile tool used for a wide range of applications by targeting specific populations of resident CNS cells to promote regeneration. 

Primary approaches to reprogram endogenous cell populations include adult non-NSPC reprogramming to generate functional neurons and NSPC reprogramming to motivate neuro- and gliogenesis. The generation of functional neurons is a minimum requirement to promote regeneration and restore signal transmission after CNS injury. However, despite successful lineage conversion into functional neurons using transcription and neurogenic factors, functional locomotor recovery is limited or non-existent [145,146]. The lack of functional recovery is likely due to insufficient number of neurons generated by the direct lineage conversion of adult non-NSPCs such as astrocytes or oligodendrocytes. In contrast, gene delivery to NSPCs stimulates endogenous neuro- and gliogenesis by activating the factors necessary to promote proliferation, differentiation, integration, or migration of target cells and progeny. Endogenous NSPC populations in the adult mammalian CNS include Nestin+ neural stem cells, NG2+ polydendrocytes, Sox2+, and ependymal progenitors [59]. Gene delivery to these populations has recently resulted in enhanced functional locomotor recovery, e.g., Gsx1 [143] and Sox2 [147].

The Cai lab has identified two neurogenic genes that bind to Notch1 locus in the role of neural differentiation in the developing spinal cord, Nkx6.1 and Gsx1 [148,149]. Studies with Nkx6.1 and Gsx1 both show a decrease in glial scar formation when they are virally overexpressed [143,150]. Nkx6.1 failed to show functional locomotor recover in mice; however, the expression of Gsx1 in ubiquitous NSPCs or Sox2 in NG2+ polydendrocyte NSPC populations resulted in an improved locomotor function after SCI and a reduction in glial scarring [143,147]. Patel et al. used a lentivirus (LV) delivery system to ectopically express the Gsx1 gene, a neurogenic transcription factor necessary for embryonic spinal cord development, in the injured spinal cord [143,151]. This resulted in the activation and enhanced proliferation of NSPCs, differentiation into subtypes of neurons, 5-HT neuronal activity, and reduced glial scarring, resulting in increased locomotor recovery, indicated by basso mouse scale (BMS) behavioral scoring. Tai et al. used an adeno-associated virus (AAV) delivery system with the NG2 promotor to ectopically express Sox2, a general neurogenic factor, in NSPC populations of NG2+ polydendrocytes, resulting in a reduction in glial scarring and increased locomotor recovery indicated by behavioral scoring [147].

The direct engineering of endogenous NSPCs may shift the cell fate of activated NSPC progeny from glial to neuronal type, resulting in an overall reduction in glial-scar-border-contributing cells. Alternatively, adult glial cell conversion into neurons may prevent some resident glial cells from contributing to the scar border but will not affect activated NSPCs and progeny. A reduction in the glial scar border accompanied by increased neuronal activity is commonly observed with a restored locomotor function after SCI [143]. NSPC-targeted gene therapy is the most feasible method to promote positive functional outcomes after SCI and uses natural injury-mediated NSPC activation to produce appropriate cell types and reduce scar formation.

The discrepancy between functional outcomes with gene therapies motivated by endogenous non-NSPC or NSPC may be attributed to a variety of reasons. Non-NSPC-based lineage reprogramming into neurons is ineffective due to a lack of newly generated cells to restore signal transmission through the neural lesion, or ineffective reprogramming efficiency into the correct cell types to maintain the excitatory/inhibitory neuron balance in the injured spinal cord [152]. The direct conversion of adult non-NSPCs results in the exact same number of cells at the injury site, whereas an NSPC-motivated therapeutic approach may result in a greater number due to enhanced proliferation [143]. Furthermore, NSPC-motivated gene therapy is effective at restoring the locomotor function because NSPCs promote the generation of more cells and the correct cell types to repopulate neural lesions via neuro- and gliogenesis, while endogenous NSPCs produce resident cells, and pose no risk for immunorejection. While both strategies reduce glial scar formation via a reduction in newly proliferated astrocytes or the direct conversion of endogenous astrocytes to other adult cells, endogenous NSPC reprogramming has led to better functional outcomes [147]. It should be noted that the ectopic gene must direct the activity of NSPCs in the injured spinal cord appropriately, thus this approach is limited by the gene of interest. 

Gene delivery is a promising method to engineer the endogenous cell activity and enhance recovery after SCI; however, the safety of viral delivery systems must be considered to develop successful clinical therapeutics. The lentiviral delivery system is a retroviral vector for the therapeutic gene, and therefore will incorporate into the host’s genome [153]. Potential negative consequences include tumorigenesis and adverse/off target effects of the cell progeny over time, ectopically expressing the therapeutic gene in the human patient. Thus, the AAV-mediated delivery system provides a more clinically safe delivery route for neurogenic or transcription factors to treat SCI [154].

Overall, the reprogramming of endogenous NSPC populations via gene therapy presents an emerging method to reduce glial scar formation and facilitate locomotor recovery in the injured spinal cord [144]. Endogenous NSPC engineering results in a greater number of cells due to proliferation and appropriate cell type development due to adult neuro- and gliogenesis. Gene delivery targeting non-NSPCs for direct lineage conversion does not result in sufficient cells to repopulate the neural lesion, promote the generation of correct neuron subtypes, or restore the excitatory/inhibitory neuron balance in the injured spinal cord [143]. Furthermore, the combination of promoter and viral serotype specificity can promote specific targeting and reprogramming of NSPC populations in the injured spinal cord, e.g., AAV5/6 with NG2 promoter to target NG2+ polydendrocytes. Through this approach, endogenous NSPCs may be stimulated to undergo neuro- and gliogenesis to generate new cells at the injury site, resulting in increased functional recovery after SCI and reduced glial scarring [147]. In the future, endogenous cell reprogramming may be the key for developing effective therapeutics for SCI and it is highly translational for broad CNS injury and degeneration.

## 3. Conclusions and Future Directions

The glial scar is inhibitory to axon growth and regeneration but is necessary to preserve viable tissue surrounding the lesion site. Cellular/molecular therapeutic strategies currently investigated for SCI regeneration target the glial scar and promote regeneration in the following ways: (1) targeting scar formation; (2) resolving the mature scar; (3) cell transplantation; and (4) endogenous cell reprogramming. Locomotor functional recovery has been observed in rodent SCI model behavioral assessments of therapeutics in each of these major groups [86,95,119,143,147]. However, few therapeutics successfully translate to clinical trials; thus, SCI remains a debilitating injury.

Clinically translational therapies may lie within the NSPC-targeted gene therapy and ECM-targeted approaches. Future potential therapeutics may also use nanoparticles for delivery or scaffolds to disrupt the inhibitory microenvironment or promote neuronal survival. Furthermore, single cell approaches are increasing our knowledge of various astrocytic subtypes, providing new genes to investigate for therapeutic application, and will fortify the cell reprogramming approach of the astrocyte response after SCI to promote regeneration. One major breakthrough in the field is the discovery of an excitatory interneuron subtype that may be responsible for stimulated walking after spinal cord injury [155]. This cell type, a result of spatiotemporal RNA sequencing, should be further investigated as a mechanism to simulate for treatment via cellular and molecular means.

## Figures and Tables

**Figure 1 cells-12-00853-f001:**
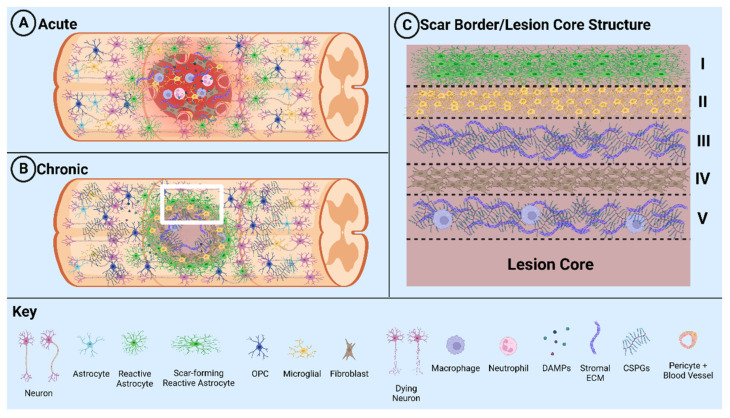
Schematic diagram illustrating glial scar formation. (**A**) Formation of the lesion and scar during the acute phase of SCI. (**B**) Formation of the fibrotic tissue and glial scar during the chronic phase of SCI. (**C**) White box region in B detailing the layers of the glial scar: I. astrocytes, II. microglia, III. secreted stromal ECM/CSPGs, IV. fibroblasts, and V. stromal ECM/CSPGs and penetrating macrophages.

**Figure 2 cells-12-00853-f002:**
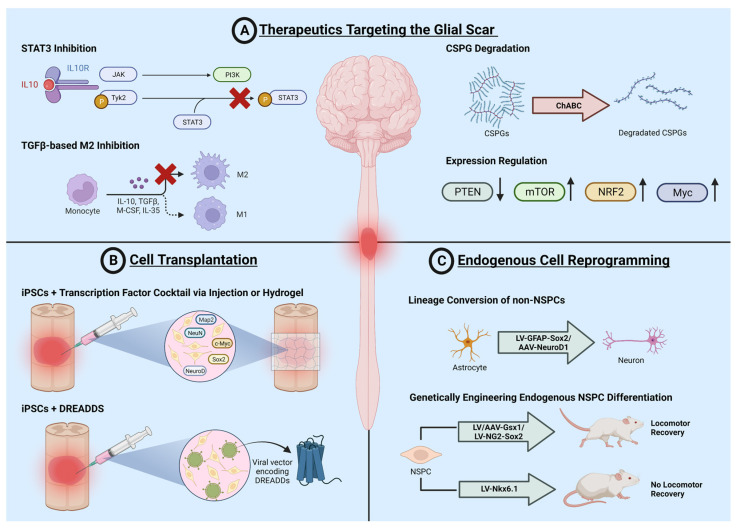
Current research in therapeutic interventions. Current cellular and molecular strategies to treat SCI include (**A**) therapeutics targeting the glial scar, (**B**) cell transplantation, and (**C**) endogenous cell reprogramming. These strategies each have their own pros and cons, with new genetic and biotechnological advances regularly changing the landscape of each type. Advances in each field over the last 5–10 years are highlighted in the following sections.

**Table 1 cells-12-00853-t001:** Positive and negative effects of the glial scar.

Positive Effects:	References:
Uptakes excess glutamate preventing chronic glutamate neurotoxicity	[70,71]
Macrophages improve overall tissue quality through removal of cell debris	[43,55,57]
Prolonged tissue repair signaling	[16,36,49]
Maintains body’s natural excitation/inhibition ratio; helps prevent irregular signaling	[45,72,73]
Physical barrier to protect remaining functional tissue	[14,29,37]
**Negative Effects:**	
Non-resolving auto-immune response that leads to fueling fibrosis; over stimulation of inflammation leads to damaged surrounding tissues	[16,74]
Over stimulation of inflammation leads to damaged surrounding tissues	[16,50,75]
Inhibits differentiation of OPCs	[26,76]
Inhibits axogenesis, plasticity of the neuron, and myelin generation	[4,5,6,13,77]
Physical barrier making transplanted/endogenous cell migration difficult	[29,35,45,78]
Excessive production of free radicals, reactive oxygen species (ROS), and glutamate, as well as ion imbalance	[14,79,80,81]

**Table 2 cells-12-00853-t002:** A list of recent clinical trials for SCI.

Study Title	Intervention	Duration	Phase	Status	Sponsor
Intrathecal Transplantation of Autologous Bone Marrow-derived Mononuclear Cells for Treating Traumatic Acute Spinal Cord Injury	Lumbar injection Transplantation of autologous bone-marrow-derived mononuclear cells	2020–2023	II	recruiting	Shanghai Changzheng Hospital
Assessment of Safety and Effectiveness of Mesenchymal Stem Cells in the Treatment of Spinal Cord Injury (SCI) Patients	Autologous bone-marrow-derived MSCs and Wharton-jelly-derived mesenchymal stem cells	2017–2020	I	completed	University of Jordan
Subarachnoid Administrations of Adults Autologous Mesenchymal Stromal Cells in SCI	Adult autologous mesenchymal bone marrow cells	2019	I	completed	Puerta de Hierro University Hospital
Intrathecal Administration (Pattern 100/3) of Expanded Autologous Adult Bone Marrow Mesenchymal Stem Cells in Established Chronic Spinal Cord Injuries	Autologous mesenchymal bone marrow cell injection	2015–2017	II	completed	Puerta de Hierro University Hospital
Autologous Bone Marrow Cell Transplantation in Persons with Acute Spinal Cord Injury—An Indian Pilot Study	Autologous mesenchymal bone marrow cells	2011–2017	I/II	completed	Indian Spinal Injuries Centre
Comparative Evaluation of Safety and Effectiveness of Autologous Bone Marrow Derived Mesenchymal Stem Cells (BM-MSC) vs. Adipose Tissue Derived Mesenchymal Stem Cells (AT-MSC) in the Treatment of Spinal Cord Injury (SCI) Patient	Intrathecal injection of autologous mesenchymal stem cells	2016–2018	I/II	completed	University of Jordan
CELLTOP Part II: A Phase II Clinical Trial of Autologous Adipose Derived Mesenchymal Stem Cells in the Treatment of Paralysis Due to Traumatic Spinal Cord Injury	Intrathecal transplantation of autologous adipose-derived mesenchymal stem cells	2020–2024	II	recruiting	Mayo Clinic
Phase I Clinical Trial of Autologous Adipose Derived Mesenchymal Stem Cells in the Treatment of Paralysis Due to Traumatic Spinal Cord Injury	Intrathecal delivery of autologous, adipose-derived mesenchymal stem cells	2017–2019	I	completed	Mohamad Bydon
Safety of Cultured Allogeneic Adult Umbilical Cord Derived Mesenchymal Stem Cells for SCI	Cultured allogeneic adult umbilical-cord-derived mesenchymal stem cells	2022–2026	I	recruiting	Foundation for Orthopaedics and Regenerative Medicine
A Randomized Controlled Phase II, Two-Arm Study of Umbilical Cord Blood Cell Transplant (MC001) Into Injured Spinal Cord Followed by the Locomotor Training for Patients with Chronic Complete Spinal Cord Injuries (SCI)	Umbilical cord blood mononuclear stem cell (UCBMSC) transplant	2022–2023	II	recruiting	StemCyte, Inc.
Allogeneic Cord Blood for Neurological Diseases in Adults	Allogeneic umbilical cord blood therapy	2022	I	Not yet recruiting	The Medical Pavilion, Bahamas
Repeated Subarachnoid Administrations of Human Umbilical Cord Mesenchymal Stem Cells in Treating Spinal Cord Injury	Intrathecal administration of human umbilical cord mesenchymal stem cells	2018–2020	I/II	completed	Limin Rong, Third Affiliated Hospital, Sun Yat-Sen University
Allogeneic Mononuclear Umbilical Cord Blood Systemic Infusions for Adult Patients with Severe Acute Contusion Spinal Cord Injury: Phase I Safety Study and Phase IIa Primary Efficiency Study	I.V. infusions of human allogeneic umbilical cord blood mononuclear cells	2013–2018	I/IIa	completed	Sklifosovsky Institute of Emergency Care
A Single Center, Open Label, Single Group, Phase 1/2a Clinical Study to Evaluate the Safety and Exploratory Efficacy of Transplantation Therapy Using PSA-NCAM(+) NPC Derived From hESC Line in AIS-A Level of Sub-acute SCI(From 7 to 60 Days)	Neural precursor cells derived from human embryonic stem cell line	2021–2023	I/IIa	recruiting	S.Biomedics Co., Ltd.
A Multi-center, Double-blind, Randomized, Placebo-controlled, Delayed Start Phase II/III Study to Assess the Efficacy and Safety of Neuro-Cells in (Sub)Acute Spinal Cord Injury Patients	Intrathecal intervention with neuro-cells	2022	II/III	recruiting	Neuroplast
A 3 Months Open Phase I Study to Assess the Safety of the Intrathecal Application of Neuro-Cells in End Stage (Chronic) Traumatic Spinal Cord Injury Patients	Intrathecal application of neuro-cells	2020–2021	I	active, not recruiting	Neuroplast
The Safety of Autologous Human Schwann Cells (ahSC) in Subjects with Chronic Spinal Cord Injury (SCI) Receiving Rehabilitation	Autologous human Schwann cell transplantation	2015–2019	I	completed	W. Dalton Dietrich, University of Miami, Miami Project
Dose Escalation Study of AST-OPC1 in Spinal Cord Injury	AST-OPC1 injection	2015–2018	I/IIa	completed	Lineage Cell Therapeutics, Inc.

Currently, only cell-therapy-based trials are available, and no gene-therapy-based trials were found; data were retrieved from ClinicalTrials.gov on 30 November 2022.

## Data Availability

All data are available from the corresponding author upon reasonable request.

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
