# Peer review of "Current Advancements in Spinal Cord Injury Research—Glial Scar Formation and Neural Regeneration"

_cells, 2023, doi:10.3390/cells12060853_

Round 1

Reviewer 1 Report

This review focuses on the cellular glial scar formation and recent advancements in cellular/molecular-based treatments to promote neural regeneration in a more favorable microenvironment. The authors discuss the cell types that take part in the formation of glial scar following spinal cord injury. Impressive and transparent figures help the reader understand the cellular events following spinal cord injury.

Although the review is thoroughly and very well written there are few points where the MS could be further improved.

   Line 85 „Reactive astrocytes secrete molecular signals that contribute to the recruitment of other cell types….” What molecular signals secreted by reactive astrocytes think the authors of? Please clarify.

  In Figure 2 C and in the „Cell transplantation” section, the iPSCs term can be read several times. These cells are embryonic-like pluripotent cells that enable the development of an unlimited source of any type of cells. Tuszynski et al. grafted iPSCs-derived neural stem cells into rats after spinal cord injury.  Others have also transplanted iPSCs-derived cells following spinal cord injury, however, there was little functional recovery in these studies. See Kobayashi et al. 2012, Nori et al. 2015, Shibata T et al. 2023. On the other hand, Bellák et al. (2020) grafted undifferentiated iPSCs into the injured rat spinal cord and found a remarkably significant functional recovery. While iPSC derivatives may be able to replace the lost cell population, undifferentiated stem cell grafts contribute to neuroprotection through a paracrine effect. The conceptual differences mentioned above should be clarified and detailed in MS.

  A vast number of ECM molecules undergo differential regulation after spinal cord injury. A short section should be inserted to introduce the various ECM components and their roles in glial scar formation.

Author Response

Response to Reviewer 1 Comments

We greatly appreciate the reviewer’s insightful and constructive comments and suggestions. We revised our manuscript accordingly. Our response and changes are highlighted in blue font.

Reviewer #1:

This review focuses on the cellular glial scar formation and recent advancements in cellular/molecular-based treatments to promote neural regeneration in a more favorable microenvironment. The authors discuss the cell types that take part in the formation of glial scar following spinal cord injury. Impressive and transparent figures help the reader understand the cellular events following spinal cord injury.

Although the review is thoroughly and very well written there are few points where the MS could be further improved.

 Line 85 „Reactive astrocytes secrete molecular signals that contribute to the recruitment of other cell types….” What molecular signals secreted by reactive astrocytes think the authors of? Please clarify.

Response: These molecules are listed in the manuscript (Line 92-93 “including yes-associated protein (YAP), CCL2, and Csf2,”). Information regarding recruitment of both astrocytes and immune populations occur in parallel with the generation of the fibrotic scar formation stage follows these markers denoting their role (Lines 93-94 “, that contribute to the recruitment of other cell types, primarily immune cells, while physically ensnaring fibroblasts by 7 days post injury (dpi)”)

In Figure 2 C and in the „Cell transplantation” section, the iPSCs term can be read several times. These cells are embryonic-like pluripotent cells that enable the development of an unlimited source of any type of cells. Tuszynski et al. grafted iPSCs-derived neural stem cells into rats after spinal cord injury.  Others have also transplanted iPSCs-derived cells following spinal cord injury, however, there was little functional recovery in these studies. See Kobayashi et al. 2012, Nori et al. 2015, Shibata T et al. 2023. On the other hand, Bellák et al. (2020) grafted undifferentiated iPSCs into the injured rat spinal cord and found a remarkably significant functional recovery. While iPSC derivatives may be able to replace the lost cell population, undifferentiated stem cell grafts contribute to neuroprotection through a paracrine effect. The conceptual differences mentioned above should be clarified and detailed in MS.

Response: We recognized the potency and unique benefits of undifferentiated stem cell grafts and provided clarity between commonly-used NSC-derived iPSC cell transplantation pools and undifferentiated non-cell fate specific grafts (Line 381-387) “The use of stem cell types non-specific to neuronal fates may provide integral qualities to induce neuroprotection via paracrine effects and regeneration. Bellak et al transplanted undifferentiated iPSCs into the rat SCI model which resulted in significant functional recovery through the generation of neuronal and other beneficial cell types. These grafts also expressed glial-derived neurotrophic factor (GDNF) and IL-10, two molecules known to possess neuroprotective qualities and enhance motor function.”).

A vast number of ECM molecules undergo differential regulation after spinal cord injury. A short section should be inserted to introduce the various ECM components and their roles in glial scar formation.

Response: We have introduced the various ECM components and their roles in glial scar formation in the Introduction and Therapeutic Strategy sections:

Line 57-61: “Myelin-associated molecules are dispersed around the injury site from injured axons and oligodendrocytes. These molecules, primarily MAG, oligodendrocyte myelin glycoprotein (OMgp), and Nogo-A, are all known inhibitors of neural, axonal regeneration, and plasticity and persist during scar formation and maturation”.

Line 95-98: “Fibroblasts and astrocytes produce extracellular matrix (ECM) molecules to stabilize the glial scar, e.g., CSPGs, fibronectin, laminin, collagen, and proteoglycans from fibroblasts, and provide shape and stability [24,27,34-36]. After 1-2 weeks, astrocytic proliferation stops, effectively representing glial scar maturation”.

Line 111-114: “The primary function of fibroblasts after SCI is to produce stromal ECM molecules and fortify the glial scar structure, as well as provide inhibitory signaling to generate a physicochemical barrier that protects the injured tissue from invading species.”. 

Line 259-263: “CSPGs form a perineural net around sprouting axons that inhibits formation in aberrant regions [73]. These nets are degraded by chondroitinase-ABC (chABC), signaling the establishment of mature neuronal circuitry [74,75]. After SCI, CSPGs are released by proliferating astrocytes and fibroblasts and inhibit local cell function to protect damaged tissue and maintain the inflammatory state”.

Thus, a dedicated ECM section may not be necessary as this information is woven into the Introduction and Therapeutic Strategy sections.

Reviewer 2 Report

The article ”Current Advancements in Spinal Cord Injury Research - Glial Scar Formation and Neural Regeneration”  focuses on cellular/molecular aspects of glial scar formation and discusses the advantages and disadvantages of strategies to promote regeneration after SCI. Improvement can take into consideration the following issues:

Major issues:

1. Presenting glial scar as a controversy of two opposing theories for defining it as positive or negative in the context of regeneration after SCI is scientifically inappropriate. As in all homeostatic balances, we have to see interlinked processes, as also others: apoptosis, inflammation, or autophagy, with dual consequences - positive or negative, depending on physiological and pathological context. So it is recommended to eliminate presentation in ”white or black,” which is contra productive for science.

2. Lack of citations, for example, the entire paragraph of the introduction - lines 30-39

3. Sections 1 to 4 (1. Introduction, 2. Cellular events immediately following trauma, 3. Glial Scar Formation, and 4. Controversy Defining the Glial Scar as Positive or Negative) are better included in one single section, Introduction,  with more consideration for the subject which must be discussed in this review article.

4. The presentation is incomplete. For example, is not discussed the role of the cellular and molecular microenvironment of the glial scar, inclusively, the ionic equilibrium.

5. It is recommended to add more recent references (2021-2023) regarding ”Current Advancements in Spinal Cord Injury Research”, considering also detrimental consequences at the cellular level, ROS, inflammation, hypoxia, and so on.

Minor issues:

1. The actual paragraph of the Introduction is very limited to few data regarding spinal cord injury and glial scars. Considering the title ”Current Advancements in Spinal Cord Injury Research - Glial 2 Scar Formation and Neural Regeneration” - it is expected to have in the Introduction a presentation of current ”state-of-art” in this scientific field of a minimum of one page, and not so few general well-known assertions.

2. It is inappropriate to publish in an international scientific journal data restricted to a regional or state area: The National Spinal Cord Injury Statistical Center estimated in 2022 that 299,000 people are living with spinal cord injuries in the United State alone, a metric that is projected to increase by 17,000 per year. (lines 33-34). It is more correct to estimate globally the situation. Otherwise, why not publish in a regional journal?

3. Traumatic brain injury (TBI) appears only in Conclusions, with no other mention in the Introduction or somewhere else. It is recommended to see the relevance of it not only in Conclusions.

Author Response

Response to Reviewer 2 Comments

We greatly appreciate the reviewer’s insightful and constructive comments and suggestions. We revised our manuscript accordingly. Our response and changes are highlighted in blue font.

Reviewer #2

The article ”Current Advancements in Spinal Cord Injury Research - Glial Scar Formation and Neural Regeneration”  focuses on cellular/molecular aspects of glial scar formation and discusses the advantages and disadvantages of strategies to promote regeneration after SCI. Improvement can take into consideration the following issues:

Major issues:

  1. Presenting glial scar as a controversy of two opposing theories for defining it as positive or negative in the context of regeneration after SCI is scientifically inappropriate. As in all homeostatic balances, we have to see interlinked processes, as also others: apoptosis, inflammation, or autophagy, with dual consequences - positive or negative, depending on physiological and pathological context. So it is recommended to eliminate presentation in ”white or black,” which is contra productive for science.

Response: We thank the reviewer for providing insight. This controversy section was not intended to provide a black or white story, but comment on the regenerative theory resulting in recent therapeutic advancements. We changed the section title from “Controversy Defining the Glial Scar as Positive or Negative” to “Positive and Negative Effects of the Glial Scar”, and removed the statement (“Thus, two opposing theories exist in the field to promote regeneration after SCI: 1. The glial scar is detrimental and should be disrupted or eliminated; and 2. The glial scar is protective and necessary to healing. The truth may lie somewhere between the extreme perspectives as the glial scar generation results in both positive and negative events after SCI.”), indicating extreme perspectives on the glial scar. We also added information to support a more balanced perspective indicating positives and negatives to each change made in therapeutic intervention (Line 184-189 “Glial scar formation results from a combination of complex processes including inflammation, reactive gliosis, apoptosis, autophagy, and others. To design effective therapeutics, the potential consequences of manipulation of each process must be carefully considered. In the field, two strategies to target the glial scar have been developed: targeting glial scar formation and targeting aspects of the established scar”)

We also stated in the conclusion that future therapeutics may require a combinatorial strategy considering interwoven positive and negative effects of the glial scar processes and aspects of the tissue/immune response (Line 193-194 “Moreover, a combinatorial approach may be most effective to establish a treatment that is optimal.”).

  1. Lack of citations, for example, the entire paragraph of the introduction - lines 30-39

Response: Citations have been added to the Introduction and other sections: Lines 34, 37,40, 186, 426, 482, 489, 496, 499, 507, 512.

  1. Sections 1 to 4 (1. Introduction, 2. Cellular events immediately following trauma, 3. Glial Scar Formation, and 4. Controversy Defining the Glial Scar as Positive or Negative) are better included in one single section, Introduction, with more consideration for the subject which must be discussed in this review article.

Response: We agree with the reviewer and have restructured our manuscript as the following:

The first four sections have combined into a single introduction under section 1 with sub sections including aspects of cellular events after injury, cellular components of the glial scar, and controversy around the glial scar (Line 44 “1.1 Cellular events immediately following trauma”, Line 80 “1.2 Glial Scar Formation”, Line 181 “1.3. Positive and Negative Effects of the Glial Scar”)

  1. The presentation is incomplete. For example, is not discussed the role of the cellular and molecular microenvironment of the glial scar, inclusively, the ionic equilibrium.

Response: We thank the reviewer for helpful comment. The goal of this manuscript is to discuss regenerative strategies to treat spinal cord injury with a focus on the glial scar as a barrier ( Line 17-24 “The scar border is a physicochemical barrier composed of elongated astrocytes, fibroblasts, and microglia secreting chondroitin sulfate proteoglycans, collogen, and dense extra-cellular matrix. While this physiological response preserves viable neural tissue, it is also detrimental to regeneration. To overcome negative outcomes associated with scar formation, therapeutic strategies have been developed: prevention of scar formation, resolution of the developed scar, cell transplantation into the lesion, and endogenous cell reprogramming. This review focuses on cellular/molecular aspects of glial scar formation, and discusses advantages and disadvantages of strategies to promote regeneration after SCI.”).

For this purpose, we provided basic formation on glial scar formation in the introduction section to facilitate a meaningful discussion of therapeutic advancements. Thus, we do not believe detailed microenvironment discussion is in the scope of this review topic and may sway the reader further from our objective.

To address your comment, we have included information about ionic imbalance after SCI (Line 46-48 “Ischemia, in addition to damage-mediated ion channel defects and rapid calcium release via cell lysis contribute to ionic imbalance at the injury epicenter”).

  1. It is recommendedto add more recent references (2021-2023) regarding ”Current Advancements in Spinal Cord Injury Research”, considering also detrimental consequences at the cellular level, ROS, inflammation, hypoxia, and so on.

Response: Recent literature has been added regarding information on ROS, hypoxia, and inflammation (Line 46, line 48, line70, line 311, line 410, and in Table 1 - line 977).

Minor issues:

  1. The actual paragraph of the Introduction is very limited to few data regarding spinal cord injury and glial scars. Considering the title ”Current Advancements in Spinal Cord Injury Research - Glial 2 Scar Formation and Neural Regeneration” - it is expected to have in the Introduction a presentation of current ”state-of-art” in this scientific field of a minimum of one page, and not so few general well-known assertions.

Response: The introduction section has been revised to present a more complete story (Line 33-42 “A recent study estimated the overall global prevalence of SCI is 20.6 million cases and 250,000 to 500,000 patients each year suffer from SCI [1]. After SCI, a tissue scar forms surrounding the injury epicenter composed of glial and supporting cell types. However, many of these invading cell types also contribute to an inflamed, inhibitory microenvironment detrimental to neural regeneration [2]. This inhibitory microenvironment suppresses neural regeneration via secreted molecules that inhibit neuronal function or prevent axogenesis, e.g. chondroitin sulfate proteoglycans (CSPGs), Nogo-A, and myelin-associated glycoprotein (MAG) [3-5]. To enhance neural regeneration, the glial scar may be manipulated to reduce its negative consequences or synergistically enhance positive qualities.”).

In its current state, this section discusses the basic elements of glial scar formation and major therapeutic types in the scope of the SCI regeneration field.

  1. It is inappropriate to publish in an international scientific journal data restricted to a regional or state area: The National Spinal Cord Injury Statistical Center estimated in 2022 that 299,000 people are living with spinal cord injuries in the United State alone, a metric that is projected to increase by 17,000 per year. (lines 33-34). It is more correct to estimate globally the situation. Otherwise, why not publish in a regional journal?

Response: The U.S. statistic has been changed to an international statistic to better reflect the journal audience (Lines 33-34 “. Recent studies estimated the overall global prevalence of SCI is 20.6 million cases and 250,000 to 500,000 patients each year suffer from SCI”).

  1. Traumatic brain injury (TBI) appears only in Conclusions, with no other mention in the Introduction or somewhere else. It is recommended to see the relevance of it not only in Conclusions.

Response: We agree with the reviewer and removed the discussion of TBI in Conclusions.

Round 2

Reviewer 2 Report

The manuscript was properly improved, and the authors responded adequately to all my comments, so I recommend it be accepted in the present form.